# Perceptual Complexity as Normalized Shannon Entropy

**DOI:** 10.3390/e27020166

**Published:** 2025-02-05

**Authors:** Norberto M. Grzywacz

**Affiliations:** 1Department of Psychology, Loyola University Chicago, Chicago, IL 60660, USA; norberto@luc.edu; Tel.: +1-773-508-2970; 2Department of Cognitive Science, Johns Hopkins University, Baltimore, MD 21218, USA

**Keywords:** perceptual complexity, Shannon Entropy, esthetic values, decision making, spatial resolution, spatial range, translational isometry

## Abstract

Complexity is one of the most important variables in how the brain performs decision making based on esthetic values. Multiple definitions of perceptual complexity have been proposed, with one of the most fruitful being the Normalized Shannon Entropy one. However, the Normalized Shannon Entropy definition has theoretical gaps that we address in this article. Focusing on visual perception, we first address whether normalization fully corrects for the effects of measurement resolution on entropy. The answer is negative, but the remaining effects are minor, and we propose alternate definitions of complexity, correcting this problem. Related to resolution, we discuss the ideal spatial range in the computation of spatial complexity. The results show that this range must be small but not too small. Furthermore, it is suggested by the analysis of this range that perceptual spatial complexity is based solely on translational isometry. Finally, we study how the complexities of distinct visual variables interact. We argue that the complexities of the variables of interest to the brain’s visual system may not interact linearly because of interclass correlation. But the interaction would be linear if the brain weighed complexities as in Kempthorne’s λ-Bayes-based compromise problem. We finish by listing several experimental tests of these theoretical ideas on complexity.

## 1. Introduction

People draw pleasure in seeing the details of a painting or listening to the variations in a musical piece. From the first known purposeful carving in Tan-Tan, Morocco [1,2], to the painted caves of Lascaux [2], to the biologically realistic paintings and sculptures of the Renaissance, artists have enjoyed adding details to their work [3,4]. The same happened in music, going from the simplicity of percussion rhythms to the complexity of Beethoven’s symphonies [5,6]. Therefore, complexity has played a key role in the arts. Complexity is not, of course, everything. In visual arts, for example, it competes with simplifying elements like symmetry, balance, and the emphasis of dramatic effects [3,7,8]. Because of competitions like this, the complexity of art tended to fall instead of increase over time in certain moments of history [9]. In addition, although people prefer stimuli that have details, incoming signals should not be overly complex [10,11,12]. Nevertheless, complexity as a source of information [4,13,14] is a major esthetic value in human endeavors.

The importance of complexity lies in the conveyance to the brain of the quantity of details, that is, the amount of information present in the stimulus [13]. The brain uses this information to allocate resources for processing the incoming sensory signal [13,15]. Hence, because of its importance, researchers have worked to try to understand how the brain computes complexity. In this process, they have developed definitions of complexity, including, for example, how many “objects” a stimulus has and perceptual scale [11,16,17,18]. Another fruitful method of defining perceptual complexity has been to use information theory [13,19] and, in particular, a normalized version of Shannon Entropy [8,13,15]. This version of complexity has been used in recent studies. These studies include the analysis of artistic [8,9,20] and architectural choices [20], psychophysical investigations of visual preferences [21], studies of the social grouping of esthetic appraisals [22], and theoretical studies of esthetic values [15,23,24].

In this article, we will focus on visual complexity for the sake of simplicity. Analyses of complexity using information theory are similar for other sensory modalities, such as, for example, in music [14,25]. The Normalized Shannon Entropy definition of visual complexity begins with the distributions of measurable variables from images. For example, luminance complexity uses the Shannon Entropy of the distribution of image luminances [8,9,15]. In turn, chromatic complexity uses the distribution of colors [15]. A more complicated type of complexity is the spatial one [8,13,20]. It uses the two-dimensional distribution of measurable values, that is, the probability that a point has a value while another point has a different value. From each of these types of distribution, one calculates the Shannon Entropy. Its normalization is then obtained by dividing by the maximum possible Shannon Entropy for an image of the same dimension and same stimulus variables (relative entropy—[26,27,28]). Thus, for example, intensity complexity requires a division by the Shannon Entropy of such an image when the intensity in each point is random among all possible values of intensities. This normalization makes the value of complexity a number from 0 to 1.

However, hypothesizing that complexity is Normalized Shannon Entropy leaves at least three theoretical open questions that one must address before testing this hypothesis experimentally. First, what are the effects of the resolution of sensory variables on complexity defined as Normalized Shannon Entropy? Although Shannon Entropy does not depend on resolution, entropy estimators do [28,29]. So, answering this question boils down to knowing whether the normalization corrects for the effects of resolution on entropy estimators. The general belief is that this correction applies [28,29], but we investigate this conviction.

Second, in a problem related to resolution, we ask what should be the spatial range in the computation of spatial complexity? The original definition considers arbitrary isometric (that is, distance-preserving) transformations of the image and asks whether two juxtaposed points predict the intensities of each other [8]. These transformations are linear combinations of translation, rotation, and reflection. Each of these transformations have in principle an infinite number of possibilities. They include the translations of distinct displacements and directions, rotations of different angles about different centers, and reflections in different orientations about different axes. But we do not know whether the brain takes all these translations, rotations, and reflections into account. For example, because we only see the surfaces of objects or regions of relative constant properties (such as the sky) in natural and urban environments [30,31,32], perhaps the only relevant translations are those at the borders between these parts of the images.

Third, how many types of complexity does the brain compute? On one hand, one can imagine that the brain computes distinct types of complexity (for example, luminance, chromatic, and spatial), keeping them separate. But if so, one must address how the brain decides on the contribution of each type to the overall esthetic value of the image. Alternatively, the brain may compute a single complexity measure before deciding on this contribution. Thus, one must understand how the brain combines different esthetic variables into a single complexity measure.

In this article, we report on a study that tries to answer these three questions. We begin this study with a general mathematical description of complexity as Normalized Shannon Entropy. We then answer these questions through mathematical analysis, use computer simulations to enrich the results, and discuss them in the context of perceptual complexity.

## 2. Theory and Results

This section has four subsections, the first supplying general definitions and each of the others addressing one of the three problems raised in Section 1. Each subsection starts with a description in plain language of the ideas and the results obtained in our analysis (“Physical Description”). Readers interested in first learning about these ideas and results without studying the mathematical details can begin by just reading these portions of this article. After each of these physical descriptions, one subsection gives the mathematical details. The results of the mathematical analysis are sometimes accompanied by computer simulations. They either illustrate the results or help answer open questions.

### 2.1. Definitions

#### 2.1.1. Physical Description

As mentioned in Section 1, we begin with the distributions of measurable image variables. Examples of these variables include luminance, color, and motion. Therefore, the measurable variable can be one-dimensional (for example, luminance) or multidimensional (for example, a vector of luminance, color, and motion). The number of values that each variable can reach can be quite large. For example, in 8-bit RGB images, the number of possible values per point is almost 17 million even without considering other measurable variables. For practical reasons, these values are often grouped in bins to reduce resolution and, thus, the total number of possibilities.

A limitation of measuring variables at each point is that this does not capture the change in complexity due to spatial organization. For example, if one scrambles the positions of the points in an image, it looks more complex, although the luminance or chromatic complexity stays the same. To quantify spatial complexity, one must perform arbitrary isometric (distance-preserving) transformations [8]. After such a transformation, one looks at the distributions of the measurable variables in pairs of juxtaposed points. These transformations are linear combinations of translation, rotation, and reflection. So, for example, for RGB images, with such a transformation, instead of looking at the distribution of triplets of values, we must look at sextuples. Each of them has one triplet before the transformation and one after it. If the image is 8-bit RGB, the number of possible sextuples is almost 300 trillion. Consequently, grouping in bins is needed even more here.

After obtaining the distribution of the measurable variables, we measure its Shannon Entropy. We then divide the outcome of this measurement by the maximum possible value of the Shannon Entropy to obtain the complexity [8] (Figure 1). The maximal value occurs when selecting the measurable values at random at each point. Because the Shannon Entropy is a non-negative number, this normalization by the maximum leads to complexity being between 0 and 1.

#### 2.1.2. Mathematical Results

Let $\vec{m}=\left(m_{1},m_{2},\cdots ,m_{N}\right)$ be an N-dimensional vector of measurable variables in a point of the image. The size of this vector can be N=1 (for example, luminance) or N>1 (for example, luminance, color, and motion). Let the set of possible values that $m_{i}$ can attain be $\left\{m_{i,1},\cdots ,m_{i,l},\cdots m_{i,L_{i}}\right\}$ (for example, in an 8-bit RGB image, $L_{i}=256$ for the red gun). Thus, the number of values that that $\vec{m}$ can attain is$$N_{\vec{m}}=\prod_{i=1}^{N}L_{i}.$$
For Image Q, let the l^th^ value among these $N_{\vec{m}}$ occur $M_{l}^{\left(Q\right)}$ times. Then, we define the probability of this l^th^ value among all $N_{\vec{m}}$ as$$P_{Q}^{\left(1\right)}\left(l\right)=\frac{M_{l}^{\left(Q\right)}}{\sum_{j=1}^{N_{\vec{m}}}M_{j}^{\left(Q\right)}}\;,$$
where the symbol “(1)” marks a Complexity of Order 1, that is, taking the measurements from one point at a time. We will consider a Complexity of Order 2 below, when we discuss spatial complexity. From this equation, we define the Entropy of Order 1 for Image Q as(1)$$H_{1}\left(Q,\vec{m}\right)=-\sum_{l=1}^{N_{\vec{m}}}P_{Q}^{\left(1\right)}\left(l\right)\log_{2}\left(P_{Q}^{\left(1\right)}\left(l\right)\right).$$
In practice, if $P_{Q}^{\left(1\right)}\left(l\right)=0$ for an l, then this term is not included in the sum, avoiding the singularity of the logarithm. This is possible because $\underset{x\to 0}{\lim}x\log x=0$.

To create an index of complexity out of this entropy, we divide it by its largest possible value given any arbitrary image. This largest entropy comes from images for which every point has a measurable value randomly picked from all possible values. Thus, $P_{Q}^{\left(1\right)}\left(l\right)=1/N_{\vec{m}}$. Substituting this result for $P_{Q}^{\left(1\right)}$ in Equation (1), one obtains the maximal Entropy of Order 1:(2)$$H_{max,1}={log}_{2}\left(N_{\vec{m}}\right).$$
Dividing Equation (1) by Equation (2) gives the Complexity of Order 1 as Normalized Shannon Entropy (Figure 1c):(3)$$C_{1}\left(Q,\vec{m}\right)=\frac{H_{1}\left(Q,\vec{m}\right)}{{log}_{2}\left(N_{\vec{m}}\right)}.\;$$
Because the denominator is $H_{max,1}$, we have $0\leq C_{1}\left(Q,\vec{m}\right)\leq 1$. We obtain 0 for single-tone images (that is, the simplest ones) and 1 for images whose measurable variables spread homogeneously through all possible values.

The Complexity of Order 2 (spatial complexity) is like the Complexity of Order 1, except that we now compare pairs of image points (Figure 1d). We begin by defining the Entropy of Order 2 for Image Q and isometric transformation T (Section 2.1.1). For this definition, we measure $M_{l_{1},l_{2}}^{\left(Q,T\right)}$. This measurement is the number of times an image point with the $l_{1}$^th^ measurement value is in juxtaposition with a point with the $l_{2}$^th^ value after the transformation. From this number, we define the following conditional probability:$$P_{Q}^{\left(2\right)}\left({l_{1},l}_{2}\left|T\right.\right)=\frac{M_{l_{1},l_{2}}^{\left(Q,T\right)}}{\sum_{j_{1}=1}^{N_{\vec{m}}}\sum_{j_{2}=1}^{N_{\vec{m}}}M_{j_{1},j_{2}}^{\left(Q,T\right)}}.$$
From this definition, we obtain the Entropy of Order 2 for Image Q and transformation T as(4)$$H_{2}\left(Q,\vec{m},T\right)=-\sum_{l_{1}=1}^{N_{\vec{m}}}\sum_{l_{2}=1}^{N_{\vec{m}}}P_{Q}^{\left(2\right)}\left({l_{1},l}_{2}\left|T\right.\right)\log_{2}\left(P_{Q}^{\left(2\right)}\left({l_{1},l}_{2}\left|T\right.\right)\right).$$

The maximal value of the Entropy of Order 2 occurs when all pairs $\left({l_{1},l}_{2}\right)$ are equally likely. Because the number of such pairs is $N_{\vec{m}}^{2}$, the maximal Entropy of Order 2 is $H_{max,2}={log}_{2}\left(N_{\vec{m}}^{2}\right)$ or(5)$$H_{max,2}={2log}_{2}\left(N_{\vec{m}}\right).$$
Dividing Equation (4) by Equation (5), one obtains the Complexity of Order 2 as Normalized Shannon Entropy (Figure 1d):(6)$$C_{2}\left(Q,\vec{m},T\right)=\frac{H_{2}\left(Q,\vec{m},T\right)}{{2log}_{2}\left(N_{\vec{m}}\right)}.$$

### 2.2. The Effects of Measurement Resolution

#### 2.2.1. Physical Description

Shannon Entropy increases with the logarithm of the number of possible measurements (Figure 2a). For example, in an 8-bit RGB image, the number of possible measurements per pixel is $2^{8\times 3}$, and therefore, if all these measurements are equally likely, then the Shannon Entropy is $\log_{2}2^{24}=24$. But for a 16-bit RGB image, the Shannon Entropy is $\log_{2}2^{16\times 3}=48$. This difference in Shannon Entropy in these two images shows that it is dependent on the resolution of measurement.

In general, the maximum possible estimated Shannon Entropy rises logarithmically with this resolution. In contrast, the maximum possible Normalized Shannon Entropy is independent of the measurement resolution, always being 1. Nevertheless, the Normalized Shannon Entropy having this upper bound does not mean that it does not change with resolution. In the next subsection, we show that as resolution rises, the Normalized Shannon Entropy also tends to, but slowly (Figure 2b). We also propose alternate related definitions of complexity that fully correct this problem of resolution.

The next subsection, however, will not address spatial resolution. What varies across images when one obtains them with the same device (or with the eye) is not the spatial locations but the measurements in each position. So, we will postpone the discussion of spatial resolution to Section 2.3.

#### 2.2.2. Mathematical Results

As shown in Equations (2) and (5), the maximum possible Shannon Entropy rises logarithmically with measurement resolution. How do non-maximal Shannon Entropies behave? To answer this question, we consider for simplicity the Complexity of Order 1 of a one-dimensional measurable variable, $m_{i}$, with probability density function, $f\left(m_{i}\right)$. Furthermore, for the sake of simplicity, we assume that the possible values of $m_{i}$ have equal spacing in the support of $f\left(m_{i}\right)$, the closed interval $\left[a,b\right]$. These values are at $m_{i,l}=\Delta m_{i}\times \left(l-1/2\right)\;,\;1\leq i\leq L_{i}$, where(7)$$\Delta m_{i}=\frac{\left(b-a\right)}{L_{i}}.\;\;$$
Thus, the Shannon Entropy is approximately$$H_{m_{i}}=-\sum_{l=1}^{L_{i}}f\left(m_{i,l}\right)\Delta m_{i}\log_{2}\left(f\left(m_{i,l}\right)\Delta m_{i}\right)\;$$
If we substitute Equation (7) for $\Delta m_{i}$ in this equation, we obtain(8)$$H_{m_{i}}=\log_{2}\left(L_{i}\right)-\left(\log_{2}\left(b-a\right)+\langle \log_{2}\left(f\left(m_{i,l}\right)\right)\rangle \right),$$
where 〈 〉 stands for “expectation” over all $m_{i,l}$. At high resolution, $\langle \log_{2}\left(f\left(m_{i,l}\right)\right)\rangle$ is approximately constant because it is the mean of $\log_{2}\left(f\left(m_{i,l}\right)\right)$ estimated with many sample points. Consequently, we asymptotically obtain(9)$$H_{m_{i}}\sim \log_{2}\left(L_{i}\right).$$
Thus, Shannon Entropy increases asymptotically with the logarithm of the measurement resolution (Figure 2a).

We can also estimate the asymptotic behavior of the Complexity of Order 1 of the measurable variable $m_{i}$. To obtain this estimate, we divide Equation (8) by Equation (2) to obtain(10)$$C_{m_{i}}\sim 1-\frac{\left(\log_{2}\left(b-a\right)+\langle \log_{2}\left(f\left(m_{i,l}\right)\right)\rangle \right)}{\log_{2}\left(L_{i}\right)}.$$
Hence, for well-distributed samples, the Normalized Shannon Entropy rises towards 1 slowly with the residual falling in proportion to the inverse of the logarithm of the resolution (Figure 2b).

Can we find alternate definitions of the Normalized Shannon Entropy whose measurement do not change with resolution? The problem with the current definition appears at high resolutions (Equations (9) and (10)). Thus, we decided to answer the question in terms of the extreme of the continuous case. The exploration of the continuous limit is akin to wanting to know what would happen if one took the human visual system as being able to measure with infite resolution. The generalization of Equation (1) for the continuous case uses the Limiting Density of Discrete Points definition [33,34,35]:(11)$$H_{1}^{\left(c\right)}\left(Q\right)=-\int d\vec{m}f_{Q}^{\left(1\right)}\left(\vec{m}\right)\log_{2}\left(\frac{f_{Q}^{\left(1\right)}\left(\vec{m}\right)}{I\left(\vec{m}\right)}\right),$$
where $f_{Q}^{\left(1\right)}$ is the probability density function associated with Image Q; I is the invariant measure; and the support for both these functions is identical, finite, and closed. I being the invariant measure means that it is the limit of the discrete samples as the resolution approaches infinity. Thus, $I\left(\vec{m}\right)$ is a probability density function.

We can use the same approach to generalize Equation (4):(12)$$H_{2}^{\left(c\right)}\left(Q,T\right)=-\int d{\vec{m}}_{1}f_{Q}^{\left(1\right)}\left({\vec{m}}_{1}\right)\int d{\vec{m}}_{2}f_{Q}^{\left(2\right)}\left(\left.{\vec{m}}_{2}\right|{\vec{m}}_{1},T\right)\log_{2}\left(\frac{f_{Q}^{\left(2\right)}\left(\left.{\vec{m}}_{2}\right|{\vec{m}}_{1},T\right)}{I\left(\vec{m}\right)}\right).$$

Because $f_{Q}^{\left(1\right)}$, $f_{Q}^{\left(2\right)}$, and I are probability density functions, an alternative interpretation of the terms after the minus sign is as a Kullback–Leibler divergence [36,37,38]. Thus, these terms are a type of statistical distance from the reference probability distribution I to either $f_{Q}^{\left(1\right)}$ or $f_{Q}^{\left(2\right)}$. This interpretation suggests that I should be the homogeneous probability density distribution. Because this distribution has the maximum possible entropy, we can characterize $f_{Q}^{\left(1\right)}$ and $f_{Q}^{\left(2\right)}$ in terms of a reduction in their entropy from the maximum.

However, Equations (11) and (12) have three problems: First, because the Kullback–Leibler divergence is positive, we obtain(13)$$H_{1}^{\left(c\right)}\left(Q\right)\leq 0\;;\;H_{2}^{\left(c\right)}\left(Q,T\right)\leq 0.$$
Since these quantities are negative, this shows that they cannot capture complexity, which is positive. Second, $H_{1}^{\left(c\right)}$ and $H_{2}^{\left(c\right)}$ are zero when $f_{Q}^{\left(1\right)}=I$ and $f_{Q}^{\left(2\right)}=I$, for example,(14)$$-\int d\vec{m}I\left(\vec{m}\right)\log_{2}\left(\frac{I\left(\vec{m}\right)}{I\left(\vec{m}\right)}\right)=0.\;$$
The quantities $H_{1}^{\left(c\right)}$ and $H_{2}^{\left(c\right)}$ should be maximized, not zero. Third, one probability density function has infinite separation from the homogenous one, namely, the Dirac delta function, $\delta \left(\vec{m}\right)$. Thus, for instance,(15)$$\underset{f_{Q}^{\left(1\right)}\left(\vec{m}\right)\to \delta \left(\vec{m}\right)}{\lim}-\int d\vec{m}f_{Q}^{\left(1\right)}\left(\vec{m}\right)\log_{2}\left(\frac{f_{Q}^{\left(1\right)}\left(\vec{m}\right)}{I\left(\vec{m}\right)}\right)=-\infty .$$
In other words, not surprisingly, in the limit of infinite resolution, the divergence from a single-tone image to a completely random one is infinity.

Despite these three problems, one can propose simple indices of complexity compatible with Equations (3) and (5). For example,(16)$$C_{1}^{\left(c\right)}\left(Q\right)=\frac{1}{1-H_{1}^{\left(c\right)}\left(Q\right)}$$
and(17)$$C_{2}^{\left(c\right)}\left(Q,T\right)=\frac{1}{1-H_{2}^{\left(c\right)}\left(Q,T\right)}.$$
These indices have all the right properties. When $f_{Q}^{\left(1\right)}=I$ or $f_{Q}^{\left(2\right)}=I$, that is, when these functions have the maximum possible entropy, we obtain $H_{1}^{\left(c\right)}=0$ or $H_{2}^{\left(c\right)}=0$ (Equation (14)), making $C_{1}^{\left(c\right)}=1$ or $C_{2}^{\left(c\right)}=1$. If in turn, $f_{Q}^{\left(1\right)}\to \delta$ or $f_{Q}^{\left(2\right)}\to \delta$, that is, if these functions have the minimum possible entropy, then we obtain $H_{1}^{\left(c\right)}\to -\infty$ or $H_{2}^{\left(c\right)}\to -\infty$ (Equation (15)), making $C_{1}^{\left(c\right)}=0$ or $C_{2}^{\left(c\right)}=0$. Finally, for any other $f_{Q}^{\left(1\right)}$ or $f_{Q}^{\left(2\right)}$, Equation (13), but not Equation (14) or (15), applies, yielding ${0<C}_{1}^{\left(c\right)}<1$ or ${0<C}_{2}^{\left(c\right)}<1$.

Other definitions for the indices of complexity are possible using the properties described in Equations (13)–(15). For example, we could conceive$$C_{1}^{\left(c\right)}\left(Q\right)=e^{H_{1}^{\left(c\right)}\left(Q\right)}.$$

Section 3 will describe experiments for testing the effects of measurement resolution on the perceived complexity.

### 2.3. The Range of Spatial Complexity

#### 2.3.1. Physical Description

So far, in this article, we defined a Complexity of Order 2 for a fixed isometric transformation (Figure 1d). Thus, we use a fixed combination of rigid translation, rotation, and reflection. However, this definition through a fixed isometric combination is not general enough to describe perceptual complexity. People can compare complexities as if they are single quantities, thus without the consideration of specific isometric combinations. To eliminate the dependence on them, Aleem et al. [8] consider alternate definitions of a Complexity of Order 2. One of these uses the mean or median complexity over all possible isometric combinations. However, this definition produces results that are not compatible with human perception. For example, if one calculates complexity for translations of different distances using natural or artistic images, it rapidly rises and then plateaus for most displacements [8] (Figure 1d). This short-rise-and-long-plateau behavior is seen because such images are composed of small subregions of relatively constant properties [30,31,32]. The problem is that the long plateau dominates the mean or median complexity but does not capture perceived complexity. Instead, perceived complexity is related to the number of subregions in the image and, thus, to the short-rise part of the complexity-versus-distance curve.

Aleem et al. also consider definitions of a Complexity of Order 2 based on short-range isometric transformations. Here, we propose to capture these solutions by defining a Complexity of Order 2 as the mean complexity in a set of small-range isometric transformations. Importantly, these ranges cannot be too small. Perceptual complexity reflects the transformations going across the boundaries between the image’s subregions. The more these transformations straddle these boundaries, the more complex an image appears to be. But the boundaries are not infinitesimally narrow, having instead a width defined by the optics of the system and the spatial resolution of the measurement (for example, discrete sampling by photoreceptors or pixels) [39,40,41].

The conclusion that the measurement of a Complexity of Order 2 uses short-range transformations has an important consequence. This type of complexity should exclude rotations and reflections. The reason for the exclusion is that they tend to cause correspondences between distant points. For rotation, if points near the center of rotation move enough to straddle boundaries and thus contribute to the computation of a Complexity of Order 2, points far from the center will move long distances. Similarly, points far from the axis of reflection will undergo large displacements. We thus propose that a Complexity of Order 2 uses only translations, not other isometric transformations. Moreover, we propose that to avoid too short and too long spatial ranges, these translations are around a small fixed distance. A perceived Complexity of Order 2 would then be the mean of the complexities for all translations with this distance. As seen in Figure 3, the complexity increases monotonically and smoothly with distance (see also Figure 1d). This increase leads us to propose that for the spatial resolution in Figure 1a,b, the fixed distance should be the smallest possible. We thus propose using a distance of 1 pixel for these figures.

#### 2.3.2. Mathematical Results

Based on the discussion of Section 2.3.1, the only relevant isometric transformation for a perceptual Complexity of Order 2 is translation. We denote this transformation as $T_{t}$. It is characterized by distance, d, and direction, θ; that is, ${T_{t}=T}_{t}\left(d,\theta \right)$. In addition, according to Section 2.3.1, the distances must be small but not too small. We thus propose that a Complexity of Order 2 independent on specific isometric transformations uses translations of $d=d_{f}$, where $d_{f}$ is a fixed distance larger than the width of boundaries. By taking the mean of Equation (5) at this distance, we obtain(18)$$C_{2}^{*}\left(Q,\vec{m}\right)=\frac{1}{2\pi }\int_{0}^{2\pi }{d\theta C}_{2}\left(Q,\vec{m},T_{t}\left(d_{f},\theta \right)\right),\;$$
where $C_{2}^{*}$ is the Complexity of Order 2 independent of specific isometric transformations. One can write equations like Equation (18) by using Equations (16) and (17) instead of Equation (5).

### 2.4. Different Types of Complexity

#### 2.4.1. Physical Description

In multiple studies using Normalized Shannon Entropy as the definition of perceptual complexity, the distinct types of complexity were kept separated. Thus, researchers have studied luminance, chromatic, and spatial complexities in isolation [8,15,20]. The visual system may treat them separately, but what happens if one sees two images and must judge which one is the more complex one? Do we perform this judgement based solely on one of these measurable variables, for example, the one with the largest complexity (Figure 1c)? Or do we combine them before the judgment? And if so, how do we combine them? One alternative is to combine them blindly (Figure 4). In this case, how much does the complexity of each variable contribute to the combination? Section 2.4.2 will show that if the variables are statistically independent, their contributions to complexity (Normalized Shannon Entropy) are proportional to their entropies (Figure 4). Alternatively, the brain may compute the complexity of the combination based on a criterion of importance, such as weighing the variables according to their contributions to survival (Figure 4). We will see next how to model all these alternatives and will discuss the different experimental predictions that each makes in Section 3.

#### 2.4.2. Mathematical Results

The complexities of individual visual variables can be obtained from Equations (3) and (5) as $C_{1}\left(Q,m_{i}\right)$ and $C_{2}\left(Q,m_{i},T\right)$, respectively. The blind method for measuring the complexities of the combination of these variables considers the full vector form of these equations. How much does the complexity of an individual variable contribute to the full complexity? If the individual variables are statistically independent, then we can rewrite an Entropy of Order 1 (Equation (1)) as$$H_{1}\left(Q,\vec{m}\right)=-\sum_{l_{1}=1}^{L_{1}}\cdots \sum_{l_{N}=1}^{L_{N}}\left(\prod_{i=1}^{N}P_{Q}^{\left(1\right)}\left(m_{i,l_{i}}\right)\right)\log_{2}\left(\prod_{i=1}^{N}P_{Q}^{\left(1\right)}\left(m_{i,l_{i}}\right)\right),$$
that is, the sum of the entropies [42,43,44]. After reorganizing the terms of this equation, we obtain$$H_{1}\left(Q,\vec{m}\right)=\sum_{i=1}^{N}H_{1}\left(Q,m_{i}\right).\;$$
Because the maximum of this entropy is the sum of the maxima, we obtain(19)$$C_{1}\left(Q,\vec{m}\right)=\frac{\sum_{i=1}^{N}H_{1}\left(Q,m_{i}\right)}{\sum_{i=1}^{N}{log}_{2}\left(L_{i}\right)\;}\;\;\;$$
by using Equations (2) and (5). Similarly,(20)$$C_{2}\left(Q,\vec{m},T\right)=\frac{\sum_{i=1}^{N}H_{2}\left(Q,m_{i},T\right)}{2\sum_{i=1}^{N}{log}_{2}\left(L_{i}\right)\;}.\;$$

Hence, the blind complexity of the combination of independent variables is proportional to the sum of their entropies (Figure 4). Thus, if a subset of the variables has entropies much larger than the others (Figure 1c), they will dominate the overall blind complexity. However, if the variables are not independent, the overall blind complexity will be smaller than the sums in Equations (19) and (20) (Figure 4). Correlation lowers entropy and thus complexity [44,45].

An alternative to using blind complexities is to consider that certain individual variables may be more important than others. Because complexities are esthetic values, they are akin to expected loss [46] or expected utility [47] in Bayesian Decision theory [48]. Consequently, we propose weighing complexities as conducted in Kempthorne’s λ-Bayes-based compromise problem [49]:(21)$$C_{1}\left(Q,\vec{m}\right)=\sum_{i=1}^{N}{\lambda_{1,i}C}_{1}\left(Q,m_{i}\right)\;$$
and(22)$$C_{2}\left(Q,\vec{m},T\right)=\sum_{i=1}^{N}{\lambda_{2,i}C}_{2}\left(Q,m_{i},T\right),\;$$
where $0\leq \lambda_{1,i},\lambda_{2,i}\leq 1$, and $\sum_{i=1}^{N}\lambda_{1,i}=\sum_{i=1}^{N}\lambda_{2,i}=1$. See Kempthorne’s theorems 3.1 and 3.2 describing the properties of the λ-based compromise problem. The weighing in Equations (21) and (22) is such that the total complexities are the means of the individual ones. One selects λ so as to emphasize the complexities for the most important variables (Figure 4).

## 3. Discussion

### 3.1. The Effects of Image Resolution

In this article, we considered the theoretical implications of using Normalized Shannon Entropy as a measure of perceptual complexity. This definition had multiple advantages, such as helping explain important properties of perceptual complexity [8,9,15,20,21,22] and being general. However, questions still remain about this definition. One is on the effects of image resolution on complexity. We showed in this article that Normalized Shannon Entropy varies systematically but little with resolution. This weak dependence settles a conjecture in the negative about whether relative entropy is constant with resolution [28,29]. One must consider this slight dependence on resolution carefully when measuring the complexity of images. But this dependence does not invalidate Normalized Shannon Entropy as a measure of perceptual complexity because the biological visual system itself has finite resolution [39,40,41]. In any event, we proposed alternate measures of complexity based on Normalized Shannon Entropy that do not have resolution dependence. These measures are based on continuous generalizations of entropy, using the Limiting Density of Discrete Points definition and Kullback–Leibler divergence [33,34,35]. These measures are not identical to the discrete versions of our definitions of complexity. Nevertheless, because efforts to obtain an empirical understanding of complexity are still ongoing, having these alternate versions help in the design of experiments to tell them apart.

### 3.2. Restrictions on Isometric Transformations Involved in Complexity of Order 2

Another open question about using Normalized Shannon Entropy as a measure for perceptual complexity has to do with a Complexity of Order 2. This is the type of complexity involved in spatial patterns [8,9,15]. The open question has to do with the proper isometric transformations to use in the computation of spatial complexity. We show that whatever the transformations are, their spatial range must be small but not too small. We then argue that this restricted range drops rotations and reflections from consideration for the computation of spatial complexity, leaving translational isometry as the only reasonable alternative. Because this must occur with a restricted spatial range, we propose the existence of a fixed measurement distance. This distance will be addressed in Section 3.5.

Rotations and reflections not being part of the computation of complexity does not mean that they do not have esthetic values. These isometric transformations create symmetries, which most people appreciate. Reflective symmetries have been found to be highly appraised in both art [3,50] and scientific studies [21,50,51,52]. Similarly, rotational symmetries are prominent in the art of diverse cultures [53,54,55].

### 3.3. How Do Different Visual Variables Contribute to Perceptual Complexity?

Yet another open question is related to the interaction between visual variables in the computation of perceptual complexity. For example, does a variable such as luminance weigh more in the computation of complexity than, say, color? Similarly, is spatial complexity more important than the luminance kind? We show that if individual variables are statistically independent, then the complexity of their combination is proportional to the sum of their entropies. Otherwise, the overall complexity is lower than predicted by this sum. Measurements from everyday images show that the latter is true; that is, interclass correlations between visual variables cause the overall complexity to be lower than expected by measuring individual variables. For example, color and luminance show correlation in natural scenes [56,57]. This correlation originates in the chemical–physical compositions of the materials in the surfaces of each scene, causing prevailing reflectance properties in terms of luminance and color [58,59].

But even if variables are independent, this does not mean that they are equally important. We thus propose that the brain may weigh distinct variables differently. Evolutionarily speaking, the luminance variable is more primal than the others [60,61]. Similarly, the space and motion variables are evolutionarily more important than color [60,61]. Therefore, we argue that luminance complexity may have a heavier weight than space and motion complexities, which, in turn, may have heavier weights than chromatic complexity (see also Section 3.6).

### 3.4. Measurements of Normalized Shannon Entropy in Natural and Human-Made Images

Complexity defined as Normalized Shannon Entropy has been measured for a variety of image types. An example is a comparison of complexities from natural (forests) with those from human-made (for example, streets, malls, and parks) settings. The former tends to yield lower complexities and broader distributions than the latter [15]. These results apply to luminance, chromatic, and spatial complexities. The only exceptions occur for the chromatic and spatial complexities of snowy rural settings. Another finding of interest in that study is that luminance complexity tends to be larger than spatial complexity (see also [8]), which tends to be larger than chromatic complexity regardless of the environment. Complexity defined as Normalized Shannon Entropy is also useful for analyzing art and architecture [8,9,20]. For example, Complexities of Order 1 and 2 are found to decrease when comparing art from the Early Renaissance with later periods [9]. Furthermore, architectural complexities are found to be larger than those of artistic paintings during the Haussmannian period in France (1853–1871—[20]). Interestingly, complexity can be different across schools of art. For example, the chromatic complexity of Impressionism is statistically significantly larger than that of other schools, such as Neoclassicism, Realism, and Romanticism. And the introduction of Chiaroscuro style in the Renaissance led to a decrease in complexity [9].

### 3.5. Other Sensory Modalities

To what extent can the conclusions of this article be generalized to other sensory modalities? The modality for which generalization is the plainest is the auditory one because the dependence on the complexity of the esthetic appraisal of auditory inputs is like that of visual ones [62,63,64]. However, the variables in audition and vision are different. In this article, we discussed variables like luminance, color, space, and motion. We must also mention the time variable as it is necessary for connecting motion to spatial change. Of these variables, only time has a direct correspondence in audition, while its other variables, for example, harmony, timbre, and rhythm, are unique to it [65,66]. Our results are easily generalizable to these variables, with little or no modifications. We thus conclude that one can capture the perceptual complexity of auditory variables with Normalized Shannon Entropy.

The situation is not as simple for the gustatory, olfactory, and somatic modalities. They have perceived complexities like those in the visual and auditory modalities [67,68,69], but their complexity properties have not been explored experimentally. Moreover, these sensory modalities have no clear organizational axes like wavelength in light or frequency in sound.

### 3.6. Experimental Predictions

No experiment to date has tested whether the properties of Normalized Shannon Entropy are applicable to the perception of complexity. However, studies have explored their connection to Shannon Entropy. For example, entropy does a good job predicting the effects of display clutter on search performance [70]. Entropy measures of complexity also account for a large amount of the variance in people’s complexity ratings of natural scenes [71]. And entropy both predicts perception uncertainty in melodic pitch [72] and accounts for the human judgment of rhythm complexity [73].

But the theoretical results in this article do lead to new predictions if one defines perceptual complexity as Normalized Shannon Entropy. We discuss four of them here:

First, one can test whether complexities due to different variables interact linearly to generate perceived complexity. For example, one can generate images by randomly assigning luminance and color to squares of multiple pixels. Thus, by adjusting the entropy of the distributions of luminance and color separately, one can create images with well-controlled complexities for each of these variables. One can then test the hypothesis that observers perceive overall complexity as a linear combination of luminance and chromatic complexities.

Second, one can use a similar technique to test whether a variable is more important than another when deciding on complexity. For example, imagine that luminance and chromatic complexities are 0.5 and 0.5, respectively, in Image A; 0.6 and 0.4 in Image B; and 0.4 and 0.6 in Image C. If luminance is more important than color for perceived complexity, then Image B will be more complex than A, which will be more complex than C. We can then use psychophysical tools to test whether participants perceive complexity following these differences. Otherwise, if a variable is more important than another, then the predicted differences will fail in favor of the principal one.

Third, the fixed distance idea for a Complexity of Order 2 can be tested. For example, imagine a high-resolution periodic image such that white vertical bars alternate with bars whose pixels have random intensities. As the spatial frequency of this image increases, so does complexity. However, if a fixed measurement distance exists, we should be able to detect it psychophysically. Complexity reaches the maximum when the distance between two white bars is equal to the fixed distance. If the spatial frequency increases further, pixels from one white bar will jump to the next white bar, lowering complexity.

Fourth, the dependence on the measurement resolution of complexity can be probed. To do so, one can ask whether people perceive a change in complexity if, for example, one changes the chromatic or luminance resolution. As shown in this article, if Normalized Shannon Entropy is the basis of perceived complexity, then it should change more slowly with resolution than entropy.

We conclude that Normalized Shannon Entropy is a practical and testable measure of perceptual complexity.

## Figures and Tables

**Figure 1 entropy-27-00166-f001:**
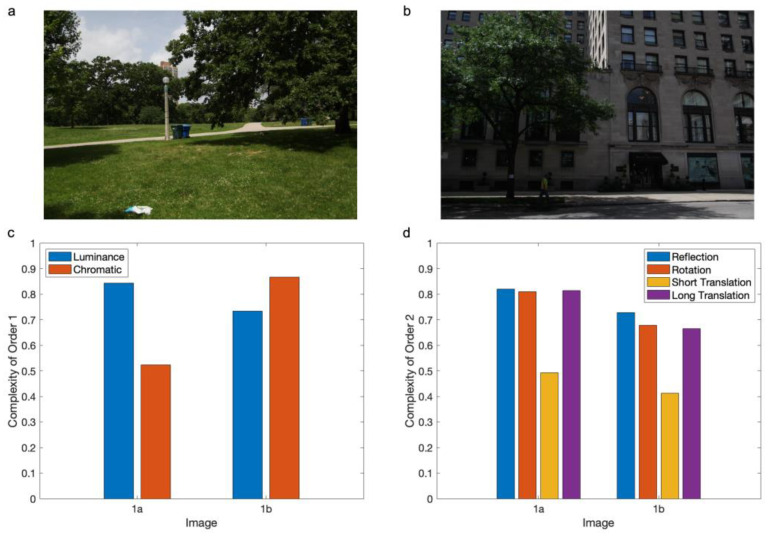
Examples of measurements of complexity as Normalized Shannon Entropy. (**a**) An image of low chromatic complexity from the dataset of Berquet et al. [15]. This image has low chromatic complexity because of the dominance of the color green across space. (**b**) An image for which the luminance complexity is lower than chromatic complexity (also from Berquet et al.). This image has lower luminance complexity because the shadows mask intensity variations. However, color provides information about, for example, the tree, thus causing larger chromatic complexity. (**c**) Complexities of Order 1 from both images. (**d**) Complexities of Order 2 for both images. A 300-pixel horizontal translational transformation leads to a high Complexity of Order 2. Similarly, horizontal reflection via an axis in the middle of the image and a 10°-anticlockwise rotation via a point in the center both yield high complexities. In contrast, a 1-pixel translational transformation leads to low complexity.

**Figure 2 entropy-27-00166-f002:**
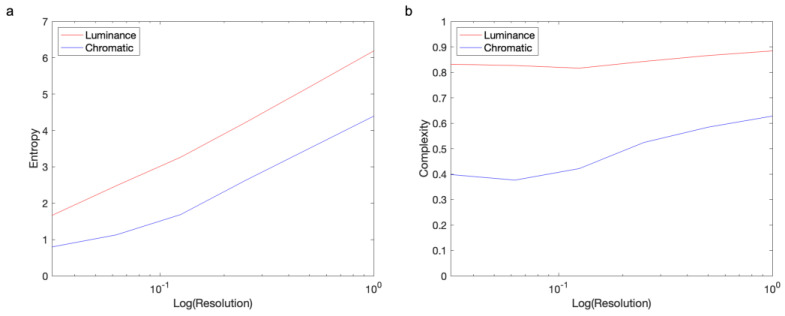
Entropy (**a**) and complexity (**b**) as a function of image resolution. These graphs are for both luminance and chromatic Complexities of Order 1, using the image in Figure 1a. The horizontal axis shows by what factor we reduced the resolution of the image. Although the entropy ascents logarithmically with resolution, the complexity as Normalized Shannon Entropy rises little.

**Figure 3 entropy-27-00166-f003:**
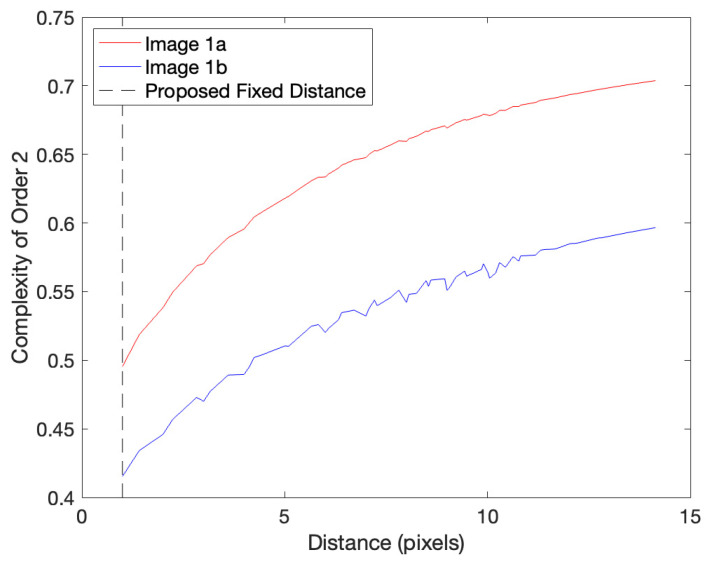
Studying candidate fixed distances for a perceived Complexity of Order 2. This figure plots the Complexity of Order 2 for the two images in Figure 1 as a function of different candidate fixed translational distances ($d_{f}$ in Equation (18)). Complexity increases monotonically and smoothly with distance (see also Figure 1d). This increase leads us to propose that for the spatial resolution in Figure 1a,b, the fixed distance should be $d_{f}=1$ pixel.

**Figure 4 entropy-27-00166-f004:**
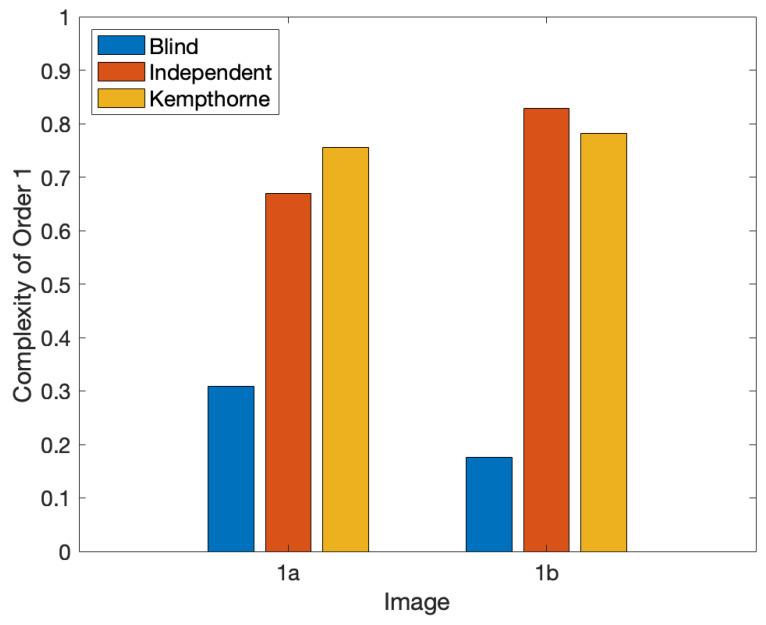
Three methods for combining complexities. This figure plots the Complexity of Order 1 for the two images in Figure 1, when combining luminance and chromatic complexities. The blind combination looks at the joint distribution of the two variables. In turn, the independent combination assumes that the luminance and color are statistically independent. Finally, the instantiation of the Kempthorne combination in this figure weighs luminance as being three times as much as color. Comparing the blind and independent combinations shows that one should not assume that different variables in images are statistically independent. Comparing the independent and Kempthorne combinations shows that the latter increases or reduces perceived complexity according to whether the more heavily weighed variable is more or less complex, respectively.

## Data Availability

The original contributions presented in this study are included in the article. Further inquiries can be directed to the corresponding author.

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
