# Peer review of "Perceptual Complexity as Normalized Shannon Entropy"

_entropy, 2025, doi:10.3390/e27020166_

Round 1

Reviewer 1 Report

Comments and Suggestions for Authors

The author presents a normalized-Shannon-entropy formalism by focusing on visual perception and discussing the effect of image resolution on complexity. It is shown that normalized Shannon entropy varies systematically but little with resolution. A study of how the complexities of distinct visual variables interact is also discussed.

The manuscript is interesting and deserves to be published in Entropy.
However, before proceeding, it is necessary to include some additional discussions that will contribute to the improvement of the manuscript, listed below:

1. The authors should clarify how the results would be impacted when analyzing an ensemble of natural images.

2. The author should provide a more detailed explanation of the subsection on experimental predictions, including results from real experiments conducted with participants.

Once the author has addressed and revised the aforementioned aspects, the manuscript will be ready for publication.

Reviewer 2 Report

Comments and Suggestions for Authors

In this work, the author analyzes the use of Normalized Shannon Entropy as a tool to quantify the perceptual complexity of images. They analyze the issues in measuring arising from resolution, and how to normalize entropy in terms of isometries for the case of spatial measures.

The manuscript per se is easy to follow and generally well-written, although some passages would benefit from further clarification (see my comments below). My only issue is that several of the points highlighted are well-known and sometimes textbook results for any reader familiar with information theory. I would suggest that the author highlight this when appropriate.

Major

1) The fact that the estimators of the Shannon entropy depend on the binning choice is a rather well-known fact. The author should make clear that the entropy of a distribution is well-defined and does not depend on the resolution; the entropy estimators, however, do. Otherwise, I am afraid that the manuscript may be confusing to the unfamiliar reader. It is also worth noting that some estimators perform better than others in this respect. Unless I am missing something, the author is using a histogram-based estimator.

2) Beginning of Section 2.2.2: I am not sure why this is considered "sampling". Sampling a probability density function f(m_i) can be done in several ways, such as via inverse transform sampling. If I understood correctly, what the author is describing is rather a form of "binning", or more precisely it is a discretization of the support of the function f.

3) Line 355: the author states "that these quantities are negative show that they are not good representatives of entropy". I believe this is very misleading since the differential entropy (the generalization of Shannon entropy to continuous distributions) can also attain negative values. It still represents an entropy. See for instance the entropy of a Gaussian distribution or the entropy of the uniform distribution between 0 and 1/2.

4) Figure 3: how does the value of d_f impact these results? Can the author comment on the choice of d_f in more in-depth?

5) Section 2.4.1 and 2.4.2: it is a textbook result that the entropy of independent events is the sum of the entropies. This should be clearly stated in the manuscript. Furthermore, I am not sure about the relation to perceptual complexity for continuous distributions. Can Eq. 19 be related in any way to Eq. 16?  My understanding is that Eq. 16 was presented as an estimator of perceptual complexity that generalizes to continuous variables, so it would be nice to see some comments about it. Finally, it would be nice to show some concrete examples where the choice of \lambda_i gives a meaningful, experimental prediction. Otherwise, Eqs. 21 and 22 is just a weighted sum.

Minor

1) Line 107: missing refernce

2) Line 191: I believe m_1 should be m_i

3) Line 257: "Thi" should be "The"

4) Line 305: in the equation it should be \Delta m_i, not \Delta m

5) Line 344: "the K-L divergence" should be "a K-L divergence"

6) Line 494: Section 2.1.4 is a wrong reference

Round 2

Reviewer 2 Report

Comments and Suggestions for Authors

I think the author did a good job replying to my concerns, and the manuscript has improved. I would just like to point out that in line 111 there is still a missing section reference, but I am sure this can be fixed during production.